# An explicit integration approach for predicting the microstructures of multicomponent alloys

Takumi Morino [1] ✉, Machiko Ode[2] & Shoichi Hirosawa [3]

Predicting the complex microstructures of practical materials has been a longstanding goal since Gibbs's pioneering work on predictions for equilibrium of heterogeneous systems. The most promising approach for achieving this goal is integrating Calculation of Phase Diagrams (CALPHAD) with phase-field models. This CALPHAD-coupled phase-field model requires two Gibbs free energy minimisation conditions: equal diffusion potential and internal equilibrium, both grounded in the second law of thermodynamics. However, as implicit functions, these minimisation conditions suffer from the curse of dimensionality when applied to multicomponent systems, which imposes significant constraints on simulation capabilities. Here we propose an approach that incorporates the equal diffusion potential and internal equilibrium conditions into a single explicit function in phase-field equations. In simulations across various practical materials, our model achieved equal diffusion and internal equilibrium conditions. Furthermore, it overcame dimensionality limitations, enabling computations for systems with up to 20 components. Thus, the proposed approach proves highly versatile and efficient, supporting a wide range of practical applications.

Many materials widely used for diverse practical applications, such as steels and super-alloys, comprise numerous elements. When developing new materials, merely optimising composition is insufficient, as material properties depend significantly on both the average composition and the microscopic texture. This microstructure is characterised by the arrangement and size of grains and phases, with differences in concentration, crystal structure, and other features. Its formation process follows the second law of thermodynamics and is theoretically treated as a Gibbs free energy minimisation process.

Microstructural control involves two key methods: the calculation of phase diagrams (CALPHAD)[1,2] and phase-field methods[3–5]. In the former method, phase diagrams are calculated to predict equilibrium states. The Gibbs free energies of the potential phases are assessed using thermodynamic models tailored to each phase, and the phase fractions and compositions are predicted as a combination that minimises the total free energy. This paper focuses on two

thermodynamic models: the quasi-regular solution model, which is primarily for liquid and solid solution phases, and the sublattice model for ordered phases, such as intermetallic compounds. In the sublattice model, the crystal lattice is divided into sublattices where specific elements are preferentially distributed. The site fractions in each sublattice are determined to minimise the free energy of the ordered phase, which is referred to as the internal equilibrium[2,6] hereinafter. CALPHAD requires two types of equilibrium calculations when a phase is modelled using the sublattice approach.

In contrast, the phase-field model simulates the process of microstructural evolution toward the equilibrium state. The system's free energy is represented using a Ginzburg–Landau-type density functional, which incorporates both Gibbs free energy and interphase and/or grain boundary energy. Then, the primary governing equation is derived by taking the functional derivative. In CALPHAD, the chemical potentials must be equal across all the equilibrium phases for a

[1]Yokohama National University, Hodogayaku, Yokohama, Japan. [2]National Institute for Materials Science, Tsukuba, Ibaraki, Japan. [3]Department of Mechanical Engineering and Materials Science, Yokohama National University, Hodogayaku, Yokohama, Japan. ✉e-mail: morino-takumi-rb@ynu.jp

multiphase system to satisfy the minimum free-energy state. The phase-field model ensures that the diffusion potentials are equal at the interfaces[7], which is equivalent to the local equilibrium concept in thermodynamics.

Adopting a Gibbs free energy function from CALPHAD is ideal for predicting the microstructural evolution of alloys[8–12]. However, the equal diffusion potential condition with the Gibbs free energy function from CALPHAD imposes significant computational-time limitations because it becomes an implicit function of the phase concentration, rendering it impractical to solve, even for ternary systems. Performing CALPHAD-coupled phase-field model calculations in a straightforward manner requires repeating the phase-diagram calculations at every computational grid point and every timestep, potentially resulting in more than a billion phase-diagram calculations.

To mitigate the computational costs of CALPHAD coupling, various strategies have been introduced, including the parabolic approximation of free-energy functions[13,14], incorporation of machine learning[15,16], and extrapolation of the equal diffusion potential condition[17–19]. However, these approaches face challenges, such as exponentially increasing complexity with additional components or coarse approximations that compromise accuracy. In the grand potential approach[20,21], the equal diffusion potential condition is automatically fulfilled; however, it is necessary to express the chemical potential explicitly as a function of composition. This involves linearising the chemical potential or using parabolic approximations of free-energy functions with respect to composition, which makes the direct use of CALPHAD functions challenging. To address this problem, we developed a phase-field model called the Direct CALPHAD Coupling (DCC) model to efficiently solve the equal diffusion potential condition without approximation[22]. However, this model becomes increasingly unstable as the number of components increases and is only applicable to quasi-regular solution-based phases.

Despite the importance of ordered phases, relevant research has been minimal owing to the substantial computational time required. Solving the internal equilibrium condition during the phase-field simulation of the superalloy γ' phase using the CALPHAD software PyCalphad[23] is estimated to take 2.3 years (see Supplementary Note 1 for details). The need to solve the internal equilibrium condition, in addition to the equal diffusion potential condition, significantly increases the computational time. Thus, there are a few examples in which the internal equilibrium condition was properly resolved during the phase-field simulation[6,24]. In most cases, either a parabolic approximation of the free energy is used[25,26] or the internal equilibrium condition is not solved at all[27–29].

The phase-field model is widely used for analysing the microstructures of materials[3–5]. However, there is no existing model capable of addressing the requirements for the development of new alloys in practice. The objective of this study is to provide a state-of-the-art method for alloy microstructure calculations by developing a phase-field model that satisfies equal diffusion potential and internal equilibrium conditions for multicomponent alloys, including liquid, solid solutions, and ordered phases, within a realistic computational timeframe.

Herein, we propose a model that incorporates equal diffusion potential and internal equilibrium conditions as an explicit function in the phase-field model. We start by redefining both conditions in the context of the phase-field model and then present a formulation incorporating them into the evolution equation of the site fraction to bypass the curse of dimensionality. The proposed model performs rapid and accurate calculations of the solidification of Al, Ni, and Fe systems and the solid-state transformation of the γ' phase—a well-known sublattice phase—for an unprecedented number of components (> 12). This is a generalised model for predicting microstructures in practical materials realised over a century after Gibbs began using thermodynamics to predict heterogeneous systems[30].

## Results

### Definition of variables and local minimisation condition

The equal diffusion potential and internal equilibrium conditions are treated separately because they are defined in the phase-field and CALPHAD methods, respectively. However, these two conditions can be expressed by a single formula[24]. In this section, we first define variables and then express the equal diffusion potential and internal equilibrium conditions as a unified equation, following Schwen's formulations[24].

In the multiphase-field model, the microstructure is represented by the phase-field variable $\phi_\alpha$. This is a function of time and position that gives the local fraction of phase $\alpha$ ($\alpha = 1, \ldots, N$ phases). For instance, $\phi_\alpha = 1$ indicates the $\alpha$ phase, whereas $0 < \phi_\alpha < 1$ represents the interface. The interfacial region, where the phase-field variable changes continuously from 0 to 1, is assumed to be a mixture of multiple phases such that $\sum_\alpha^N \phi_\alpha = 1$. At the interface, the phase composition fulfils the following mixture rule:

$$c^i = \sum_\alpha^N \phi_\alpha c_\alpha^i, \tag{1}$$

where $c^i$ represents the overall composition or mixture composition (solvent $i = n$ and solute $i = 1, \ldots, n-1$) and $c_\alpha^i$ represents the composition of component $i$ of phase $\alpha$. The relationship between the phase composition and site fraction is defined as follows:

$$c_\alpha^i = \sum_{a \in \alpha} S_a y_a^i, \tag{2}$$

where $S_a$ is the stoichiometric coefficient representing the ratio of the number of atomic positions belonging to sublattice $a$ to the total number of atomic positions in phase $\alpha$ such that $\sum_{a \in \alpha} S_a = 1$ and $y_a^i$ is the site fraction of component $i$ of sublattice $a$ in phase $\alpha$ such that $\sum_i^n y_a^i = 1$. $a \in \alpha$ implies that the summation is taken only for the phase $\alpha$. In the quasi-regular solution model, the crystal lattice is not divided into sublattices, i.e., $S_a = 1$; thus, the site fraction is equal to the phase composition. The condition that minimises the local Gibbs free energy of the microstructure while satisfying Eqs. (1) and (2) is derived using Lagrange's multiplier[24] (see Supplementary Note 2 for details):

$$\frac{1}{S_a}\frac{\partial f_\alpha}{\partial y_a^i} = \frac{1}{S_b}\frac{\partial f_\beta}{\partial y_b^i}, \tag{3}$$

where $f_\alpha$ represents the chemical free energy density of phase $\alpha$. The right and left sides of Eq. (3) correspond to the diffusion potential of each phase, as follows[24]:

$$\frac{1}{S_a}\frac{\partial f_\alpha}{\partial y_a^i} = \frac{1}{\partial c_\alpha^i/\partial y_a^i}\frac{\partial f_\alpha}{\partial y_a^i} = \frac{\partial f_\alpha}{\partial c_\alpha^i}. \tag{4}$$

Therefore, Eq. (3) represents the equal diffusion potential condition for the interface where $\alpha \neq \beta$. When $\alpha = \beta$ and $\alpha$ is described by the sublattice model, Eq. (3) becomes a condition that minimises the Gibbs free energy of a phase, as described by the sublattice model, with the constraint of a fixed phase composition. This is the exact definition of the internal equilibrium condition. In this study, Eq. (3), which encompasses the equal diffusion potential and internal equilibrium conditions, is referred to as the local minimisation condition because it is realised by minimising the local Gibbs free energy of the microstructure.

## Explicitly solvable local minimisation condition

In conventional simulations, the site fraction is treated as a dependent variable on the composition and phase-field variables. However, the time evolution of the composition is essentially driven by changes in

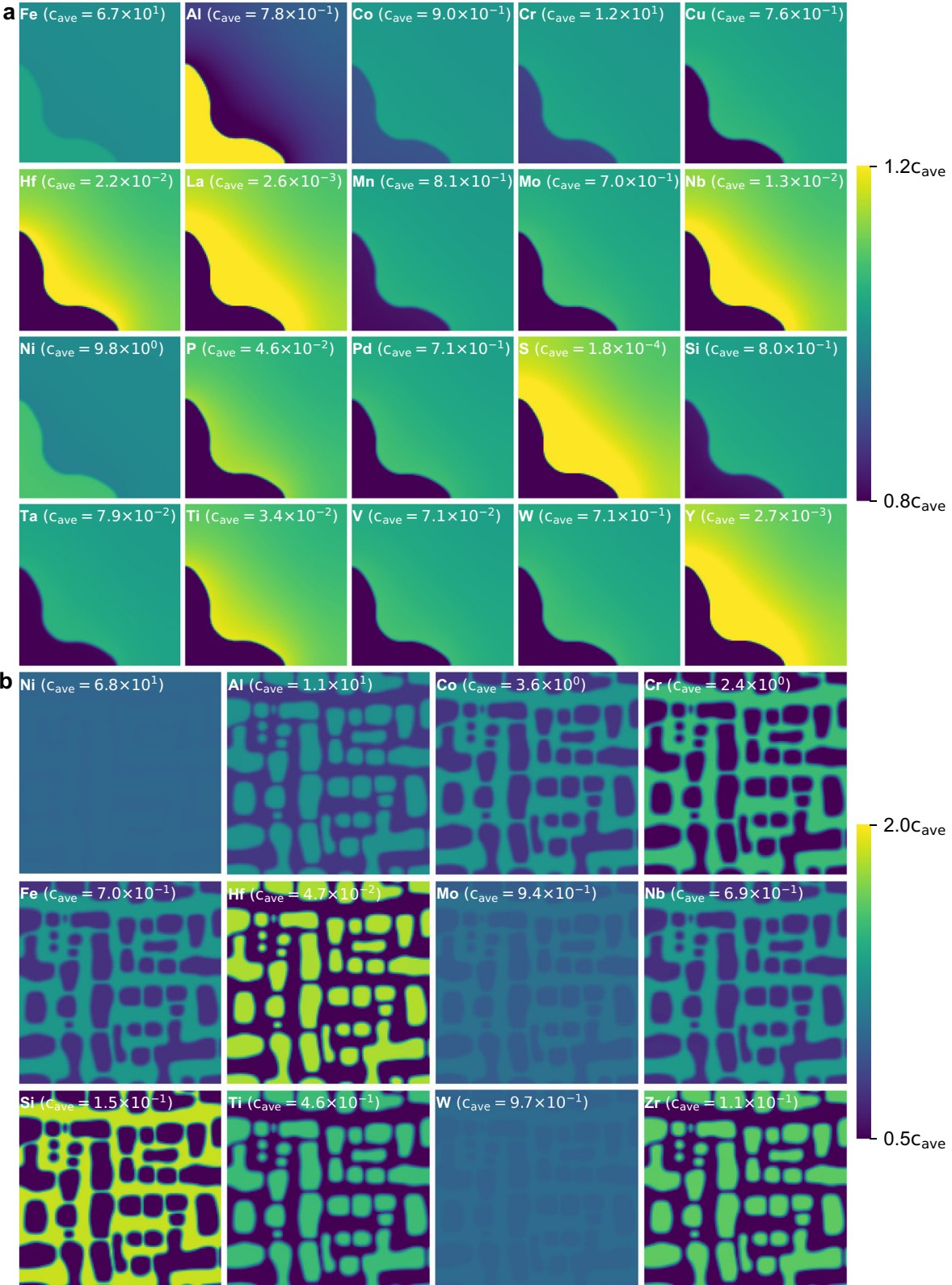

**Fig. 1 | Snapshots of composition distributions at the end of the two simulations. a** Solidification of an fcc phase in a 20-component Fe system at a solidification time of $6.0 \times 10^{-2}$ s. **b** $\gamma'$ phase precipitation and coarsening in a 12-component Ni system at an ageing time of $1.5 \times 10^3$ s. The range of the colourmaps is based on the average composition of each element ($c_{ave}$). Owing to the direct use of the CALPHAD database, the partitioning of each element varies according to the phase diagram. The CALPHAD database is obtained from the MatCalc open database (https://www.matcalc.at/).

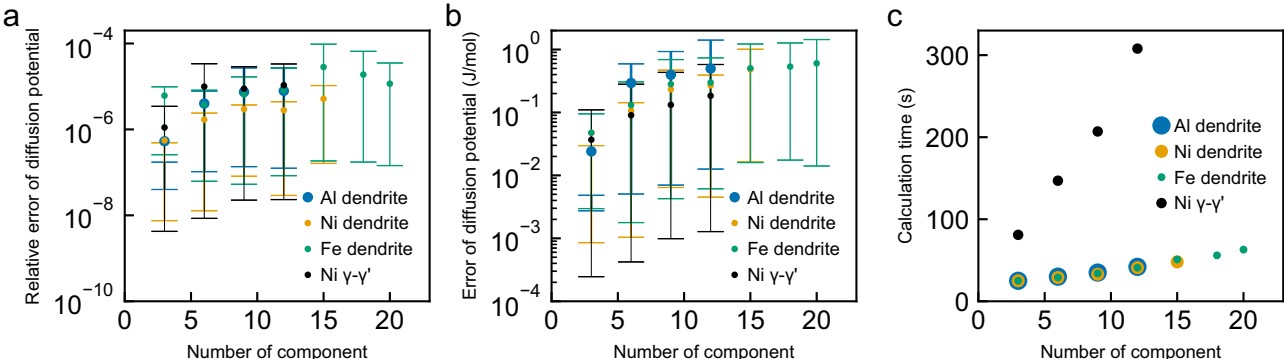

**Fig. 2 | Error of the local minimisation condition and calculation time for different numbers of components. a** Average absolute relative error of the diffusion potential. **b** Average absolute discrepancy of the diffusion potential. **c** Calculation time. The errors were calculated every 100 timesteps in regions where multiple phases and/or sublattices coexist and subsequently averaged. Blue, orange, and green markers represent the simulation results of dendritic solidification in Al-, Ni-, and Fe- systems, respectively, as case studies using the quasi-regular solution model, which requires the equal diffusion potential condition. Black markers indicate the results of $\gamma$-$\gamma'$ solid-state transformation calculated with the sublattice model, which requires both the equal diffusion potential condition and the internal equilibrium condition. In panels (**a**) and (**b**), error bars show the 95th and 5th percentiles of the computed values. Source data are provided as a Source Data file.

the site fraction, which moves toward the local minimisation condition because the composition is expressed by the summation of the site fractions, as given by Eqs. (1) and (2). Therefore, explicit treatment of the site fraction is a more reasonable approach. In this study, we formulated the evolution equation of the site fraction in the form of an explicit function that moves toward the local minimisation condition and is consistent with the conventional evolution equations of the phase-field variable and composition. The evolution equation presented next is derived in the Supplementary Notes 3 and 4.

In the context of the Euler method, changes in the site fraction per step for sublattices $a$ and $b$ are specified as $\Delta y_a^i$ and $\Delta y_b^i$, respectively, and their ratios are denoted as $k_{ba}^i$. We define the diffusion potential difference function $H^i$, which corresponds to the difference between the right and left sides of Eq. (3) after changes in the site fraction, as follows:

$$H^i\left(\Delta y_a^1, \ldots, \Delta y_a^{n-1}\right) = \frac{1}{S_a}\frac{\partial f_\alpha}{\partial y_a^i}\left(\Delta y_a^1, \ldots, \Delta y_a^{n-1}\right) - \frac{1}{S_b}\frac{\partial f_\beta}{\partial y_b^i}\left(k_{ba}^1\Delta y_a^1, \ldots, k_{ba}^{n-1}\Delta y_a^{n-1}\right). \quad (5)$$

We apply mathematical processing to Eq. (5), including expansion to the first order, application of a condition for ensuring numerical stability, and solution of the local minimisation condition ($H^i = 0$). This process produces the $k_{ba}^i$ that results in the $\Delta y_a^i$ and $\Delta y_b^i$ moving forward the local minimisation condition, as follows (see Supplementary Notes 3 and 4 for details):

$$k_{ba} = \frac{\frac{1}{S_a}\frac{\partial f_\alpha}{\partial y_a^i} - \frac{1}{S_b}\frac{\partial f_\beta}{\partial y_b^i} + (n-1)\Delta y_a^i \frac{1}{S_a}\frac{\partial^2 f_\alpha}{(\partial y_a^i)^2}}{(n-1)\Delta y_a^i \frac{1}{S_b}\frac{\partial^2 f_\beta}{(\partial y_b^i)^2}}. \quad (6)$$

By combining Eq. (6) with two necessary conditions for the phase-field model, including solute diffusion in bulk phases and conservation of solute around the moving interface, we can obtain the $\Delta y_a^i$ that moves forward to the local minimisation condition as follows (see Supplementary Note 3 for details):

$$\Delta y_a^i = \frac{-\sum_{\beta=1}^N \left(\phi_\beta + \frac{\partial \phi_\beta}{\partial t}\Delta t\right)\sum_{b\in\beta} S_b \left(\frac{1}{S_a}\frac{\partial f_\alpha}{\partial y_a^i} - \frac{1}{S_b}\frac{\partial f_\beta}{\partial y_b^i}\right) / \left(\frac{n-1}{S_b}\frac{\partial^2 f_\beta}{(\partial y_b^i)^2}\right) + \left(\frac{\partial c^i}{\partial t} - \sum_{\beta=1}^N \frac{\partial \phi_\beta}{\partial t}\sum_{b\in\beta} S_b y_b^i\right)\Delta t}{\sum_{\beta=1}^N \left(\phi_\beta + \frac{\partial \phi_\beta}{\partial t}\Delta t\right)\sum_{b\in\beta}\frac{S_b^2}{S_a}\frac{\partial^2 f_\alpha}{(\partial y_a^i)^2} / \frac{\partial^2 f_\beta}{(\partial y_b^i)^2}}. \quad (7)$$

Solving Eq. (7) is straightforward because all variables on the right-hand side are known, making it an explicit function. The first term in the numerator of Eq. (7) adjusts the site fraction to meet the local

minimisation condition. As a result, solving Eq. (7) alone satisfies the local minimisation condition without the need for convergence calculations, which is expected to significantly reduce the computational cost. This equation can be reduced analytically to obtain the standard evolution equation for compositions (see Supplementary Note 5 for details). Accordingly, an arbitrary evolution equation for compositions can be employed. In the solidification simulations presented in the following section, an anti-trapping current-infused diffusion equation necessary for quantitative simulations[31] is employed.

In the present model, the local minimisation condition and free boundary problems[32], including solute diffusion in the bulk phases, conservation of solute around the moving interface, and the Gibbs–Thomson effect, are solved using Eq. (7) and an evolution equation of the phase-field variable of the standard multiphase-field model[33,34]. Furthermore, in the calculation of $\gamma$-$\gamma'$, the mechanical equilibrium condition is solved to account for the effect of the elastic field[35].

## Case study

Two types of microstructural evolution problems are addressed to validate the proposed model. The first problem involves the solidification of the solid solution phase described by the quasi-regular solution model, and the second problem involves the solid-state transformation of the $\gamma'$ phase using the sublattice model. Note that the proposed model directly employs CALPHAD functions without the need for modifications, such as linearising the chemical potential or applying parabolic approximations to free-energy functions. This approach allows efficient computation regardless of temporal and spatial temperature changes, including those occurring during processes such as alloy solidification.

The computational process began with (i) setting the initial conditions for temperature, composition, and phase fields. The initial site fraction was input as the equilibrium values corresponding to these conditions. Subsequently, (ii) the governing equations of the phase-field variable and composition are solved. Following this, (iii) the governing equation of site fraction represented by Eq. (7) is addressed. It should be noted that the phase compositions are automatically determined through their relationship with the site fraction, as described in Eq. (2). Finally, (iv) steps (ii) and (iii) are iteratively repeated. All CALPHAD data, including those for the quasi-regular solution model and the sublattice model, have been implemented directly in the phase-field simulation code. For ultra-multicomponent systems, such as those with 20 components, manually hard-coding becomes impractical due to complexity; therefore, we developed an automatic

**Table 1 | Numerical parameters**

| | Liquid-fcc Al system | Liquid-fcc Ni system | Liquid-fcc Fe system | $\gamma$-$\gamma'$ Ni system |
|---|---|---|---|---|
| Diffusivity in liquid (m²/s) | $1.0 \times 10^{-9}$ | $1.0 \times 10^{-9}$ | $1.0 \times 10^{-9}$ | – |
| Diffusivity in solid (m²/s) | $1.0 \times 10^{-13}$ | $1.0 \times 10^{-13}$ | $1.0 \times 10^{-13}$ | $1.0 \times 10^{-16}$ |
| Interface energy (J/m²) | 0.2 | 0.2 | 0.2 | 0.02 |
| Anti-phase boundary energy (J/m²) | – | – | – | 0.05 |
| Molar volume (m³/mol) | $1.0 \times 10^{-5}$ | $1.0 \times 10^{-5}$ | $1.0 \times 10^{-5}$ | $1.0 \times 10^{-5}$ |
| Strength anisotropy (-) | 0.03 | 0.03 | 0.03 | – |
| Interface mobility (s mol/J) | $\infty$ | $\infty$ | $\infty$ | $4.1 \times 10^{-17}$ |
| Elastic constant $C_{11}$ (GPa) | – | – | – | 250.8 |
| Elastic constant $C_{12}$ (GPa) | – | – | – | 150.0 |
| Elastic constant $C_{44}$ (GPa) | – | – | – | 123.5 |
| Lattice misfit between $\gamma$ and $\gamma'$ (-) | – | – | – | 0.006 |
| Initial temperature (K) | 900 | 1700 | 1700 | 1400 |
| Cooling rate (K/s) | 50 | 50 | 50 | 0 |
| Grid resolution (m) | $1 \times 10^{-7}$ | $1 \times 10^{-7}$ | $1 \times 10^{-7}$ | $5 \times 10^{-9}$ |
| Interface thickness | $4 \times 10^{-7}$ | $4 \times 10^{-7}$ | $4 \times 10^{-7}$ | $2 \times 10^{-8}$ |
| Number of grids (-) | 128 × 128 | 128 × 128 | 128 × 128 | 128 × 128 |
| Number of timesteps (-) | $3 \times 10^{4}$ | $3 \times 10^{4}$ | $3 \times 10^{4}$ | $3 \times 10^{4}$ |
| Discrete time width (s) | $2 \times 10^{-6}$ | $2 \times 10^{-6}$ | $2 \times 10^{-6}$ | $5 \times 10^{-2}$ |
| Boundary condition (-) | Zero Neumann | Zero Neumann | Zero Neumann | Periodic |

code generation system as described in the Method section. In the case of commercial, encrypted databases, the necessary values for calculations can be obtained through the software's API.

Placing a seed of the solid solution phase in the liquid phase and cooling it enabled the growth of dendrites, as shown in Fig. 1a. When the $\gamma'$ phase is randomly distributed within the $\gamma$ phase and subjected to isothermal aging, it grows as depicted in Fig. 1b. In Fig. 1b, the $\gamma'$ phase exhibits a cuboidal morphology owing to the contribution of elastic energy. As these simulations directly utilise the CALPHAD database without simplifications for computational speed, Fig. 1 shows the partitioning of each element into phases based on a phase diagram, ensuring accurate and reliable results. Solidification and solid-state transformation calculations were performed for various numbers of components. However, the figures for these systems are omitted here, as they exhibit similar patterns of concentration distribution, confirming the robustness of our computational method across different alloy compositions.

The error in the local minimisation condition was defined as the error between the right and left sides of Eq. (3), that is, the error in the diffusion potential, at every 100 timesteps in the regions where multiple phases and/or sublattices coexist. Figure 2a depicts the average absolute relative error of the diffusion potential. Regardless of the number of components, the system, and the CALPHAD model, the error is of the order $10^{-5}$. This is comparable to the precision of the single-precision floating-point number type, which is of the order $10^{-6}$. Fig. 2b shows the average absolute discrepancy in the diffusion potential. The discrepancy is of the order $10^{-1}$ J/mol under all conditions. This value is negligibly small compared to the $10^{2}$–$10^{3}$ J/mol deviation in the diffusion potential caused by the curvature effect, which was calculated using Equation (A5) from ref. 36. Therefore, it was confirmed that the local minimisation condition is maintained throughout the simulation.

Fig. 2c shows the calculation time for different numbers of components. The calculation time for the solid-state transformation of the $\gamma'$ phase is longer than that for the solidification of the solid solution phase. This is attributed to differences in the thermodynamic models. In the solid solution phase described by the quasi-regular solution

model, thermodynamic functions, including $\frac{\partial f_\alpha}{\partial y_a^i}$ and $\frac{\partial^2 f_\alpha}{(\partial y_a^i)^2}$, in Eq. (7) have to be computed only at the interface to satisfy the equal diffusion potential condition, whereas in the $\gamma'$ phase using the sublattice model, thermodynamic functions must also be computed in the bulk to fulfil the internal equilibrium condition. Nevertheless, the calculation of the sublattice model is sufficiently fast to complete a simulation with 12 components in just 293 s. The calculation time increased linearly—not exponentially—with an increase in the number of components, indicating that our model overcomes the curse of dimensionality. Therefore, a simulation with an unparalleled number of components (20) was completed in 63 s.

Our model is superior to conventional methods with regard to both the number of components and accuracy. Conventional CALPHAD-coupled phase-field simulations include three components utilising machine learning[15], three components employing parabolic approximation[37], and ten components using the extrapolation method[18]. However, all these methods use numerical techniques to circumvent the equal diffusion potential or internal equilibrium conditions and thus inevitably compromise the accuracy of the computation. The Newton-based method can be used to solve the conditions directly; however, it is computationally expensive. The existing Newton solver for the equal diffusion potential condition—Thermo4PFM—supports only two or three components[38]. In contrast to these methods, our model changes the governing equations rather than relying on numerical techniques. Consequently, calculations involving up to 20 components—significantly exceeding the number of components for conventional methods—were achieved with high accuracy and speed without approximation or additional computational parameters.

In this study, we developed a model for incorporating equal diffusion potential and internal equilibrium conditions into an explicit function that solves the problems of solute diffusion in bulk phases and the conservation of the solute around the moving interface. The proposed approach can be directly integrated with the CALPHAD database without modifying functional forms. The computational accuracy and efficiency of our model were evaluated by performing

## Table 2 | Compositions of the liquid-fcc Al system

| System | Composition (mol%) |
|---|---|
| Ternary | Al-0.5Cu-6.3Mg |
| Senary | Al-3.8Cu-0.5Mg-0.5Cr-0.5Fe-0.5Mn |
| Nonary | Al-2.8Cu-0.5Mg-0.5Cr-0.5Fe-0.5Mn-0.5Ni-0.5Sc-0.5Si |
| Duodecimal | Al-3.6Cu-0.5Mg-0.5Cr-0.5Fe-0.5Mn-0.5Ni-0.5Sc-0.5Si-0.5Ti-0.5Zn-0.5Zr |

## Table 3 | Compositions of the liquid-fcc Ni system

| System | Composition (mol%) |
|---|---|
| Ternary | Ni-16.3Al-1Co |
| Senary | Ni-13.6Al-1Co-1Cr-0.3Cu-1Fe |
| Nonary | Ni-11.6Al-1Co-1Cr-0.3Cu-1Fe-0.3Hf-0.3Mn-1Mo |
| Duodecimal | Ni-5.2Al-1Co-1Cr-0.3Cu-1Fe-0.3Hf-0.3Mn-1Mo-1Nb-1Si-1Ti |
| Quindecimal | Ni-4.5Al-1Co-1Cr-0.3Cu-1Fe-0.3Hf-0.3Mn-1Mo-1Nb-1Si-1Ti-0.3V-1W-0.3Zr |

## Table 4 | Compositions of the liquid-fcc Fe system

| System | Composition (mol%) |
|---|---|
| Ternary | Fe-22.6Cr-10Ni |
| Senary | Fe-26Cr-10Ni-1Al-1Co-1Cu |
| Nonary | Fe-23.5Cr-10Ni-1Al-1Co-1Cu-0.1Hf-0.1La-1Mn |
| Duodecimal | Fe-21Cr-10Ni-1Al-1Co-1Cu-0.1Hf-0.1La-1Mn-1Mo-0.1Nb-0.1P |
| Quindecimal | Fe-16.5Cr-10Ni-1Al-1Co-1Cu-0.1Hf-0.1La-1Mn-1Mo-0.1Nb-0.1P-1Pd-0.1S-1Si |
| Octodecimal | Fe-16Cr-10Ni-1Al-1Co-1Cu-0.1Hf-0.1La-1Mn-1Mo-0.1Nb-0.1P-1Pd-0.1S-1Si-0.1Ta-0.1Ti-0.1V |
| Vigesimal | Fe-13.5Cr-10Ni-1Al-1Co-1Cu-0.1Hf-0.1La-1Mn-1Mo-0.1Nb-0.1P-1Pd-0.1S-1Si-0.1Ta-0.1Ti-0.1V-1W-0.1Y |

## Table 5 | Compositions of the γ-γ′ Ni system

| System | Composition (mol%) |
|---|---|
| Ternary | Ni-18.9Al-5Co |
| Senary | Ni-16.9Al-5Co-5Cr-1Fe-0.3Hf |
| Nonary | Ni-17Al-5Co-5Cr-1Fe-0.3Hf-1Mo-1Nb-1Si |
| Duodecimal | Ni-14.5Al-5Co-5Cr-1Fe-0.3Hf-1Mo-1Nb-1Si-1Ti-1W-0.3Zr |

numerical tests under various conditions, including quasi-regular solution and sublattice models; Al, Ni, and Fe systems; and 3–20 components. Regardless of the CALPHAD model, system, and number of components, the local minimisation condition, which is a unified expression of the equal diffusion potential and internal equilibrium conditions, was sufficiently fulfilled. Moreover, an unprecedented calculation of a 20-component system was performed in just 1 min using a standard personal computer. The proposed model enables the simulation of the microstructural evolution of multicomponent practical materials without the curse of dimensionality.

The proposed model is not implementable for certain specific and less frequently used CALPHAD models. Future work will focus on extending our model to encompass all CALPHAD models.

## Methods

### Simulation conditions

The time evolution equations are discretised using a finite difference method: the first-order Euler finite difference scheme is used for time

integration, and the second-order central finite difference scheme is used for space discretisation. All the simulations are conducted using a double-precision floating-point number. The numerical parameters for each calculation system are presented in Table 1. The compositions of the liquid-fcc Al system, liquid-fcc Ni system, liquid-fcc Fe system, and γ-γ′ Ni system are presented in Tables 2–5.

### Governing equations of phase, composition, and elastic fields

The following multiphase-field equation is used as the governing equation for the phase-field variable:

$$\frac{\partial \phi_\alpha}{\partial t} = -\frac{2}{N}\sum_{\beta=1}^{N} M_{\alpha\beta} \left\{ \begin{array}{l} \sum_{\gamma=1}^{N}\left[\left(W_{\alpha\gamma}-W_{\beta\gamma}\right)\phi_\gamma + \frac{1}{2}\left(a_{\alpha\gamma}^2 - a_{\beta\gamma}^2\right)\nabla^2\phi_\gamma\right] \\ + \frac{8}{\pi}\sqrt{\phi_\alpha\phi_\beta}\left[f_\alpha - f_\beta - \sum_{i=1}^{n-1}\frac{\partial f_\alpha}{\partial c_\alpha^i}\left(c_\alpha^i - c_\beta^i\right) + \Delta G_{\alpha\beta}^{elastic}\right] \end{array} \right\},$$

(8)

where $\phi_\alpha$ is the phase-field variable, $N$ represents the number of phases, $M_{\alpha\beta}$ represents the phase-field mobility of the phase field between phases $\alpha$ and $\beta$, $W_{\alpha\beta}$ represents the height of double-obstacle potential between phases $\alpha$ and $\beta$, $a_{\alpha\beta}$ is the gradient coefficient between phases $\alpha$ and $\beta$, $f_\alpha$ represents the chemical free energy density of phase $\alpha$, $c_\alpha^i$ represents the phase composition of component $i$ of phase $\alpha$, and $\Delta G_{\alpha\beta}^{elastic}$ represents the elastic driving force. The driving force for interface motion due to chemical free energy is calculated from the distance between the parallel tangents to the free-energy curves of the two phases, assuming that the interface is in quasi-equilibrium[7,17]. Therefore, the present model is not applicable to cases where the quasi-equilibrium condition is violated, such as in mixed-mode transformations. $M_{\alpha\beta}$, $W_{\alpha\beta}$, and $a_{\alpha\beta}$ are expressed as follows:

$$M_{\alpha\beta} = \frac{\pi^2}{4\delta} m_{\alpha\beta}$$

(9)

$$a_{\alpha\beta} = \sqrt{2\delta_{\alpha\beta}\sigma_{\alpha\beta}}$$

(10)

$$W_{\alpha\beta} = 4\frac{\sigma_{\alpha\beta}}{\delta_{\alpha\beta}},$$

(11)

where $m_{\alpha\beta}$, $\delta_{\alpha\beta}$, and $\sigma_{\alpha\beta}$ represent the interface mobility, interface thickness, and interface energy between phases $\alpha$ and $\beta$, respectively.

According to the standard multiphase-field model, the governing equation of the composition field in Eq. (7) is given as follows[17,39,40]:

$$\frac{\partial c^i}{\partial t} = \nabla \cdot \sum_{\alpha=1}^{N} \phi_\alpha D_\alpha^i \nabla c_\alpha^i,$$

(12)

where $D_\alpha^i$ is the diffusion coefficient of composition $i$ in phase $\alpha$. Solidification is calculated by introducing the anti-trapping current required for quantitative simulation[41] to the governing equation of the composition field:

$$\frac{\partial c^i}{\partial t} = \nabla \cdot \sum_{\alpha=1}^{N} \phi_\alpha D_\alpha^i \nabla c_\alpha^i + \nabla \cdot \sum_{\alpha>\beta}^{N} \sum_{\beta=1}^{N} J_{\alpha\beta}^i$$

(13)

$$J_{\alpha\beta}^i = \frac{a_{\alpha\beta}}{\sqrt{2W_{\alpha\beta}}}\left(c_\alpha^i - c_\beta^i\right)\sqrt{\phi_\alpha\phi_\beta}\frac{\partial \phi_\alpha}{\partial t}\frac{\nabla\left(\phi_\alpha - \phi_\beta\right)}{\left|\nabla\left(\phi_\alpha - \phi_\beta\right)\right|},$$

(14)

where $J_{\alpha\beta}^i$ represents the anti-trapping current of component $i$. This equation is derived assuming no diffusion in phase $\beta$. When the anti-trapping current is used, the relationship between the phase field and

interface mobilities is derived under the thin-interface limit condition[42]:

$$\frac{1}{m_{\alpha\beta}} = \frac{\sigma_{\alpha\beta}}{M_{\alpha\beta}a_{\alpha\beta}^2} - \frac{\pi}{8}\frac{a_{\alpha\beta}}{\sqrt{2W_{\alpha\beta}}}\zeta_{\alpha\beta}, \quad (15)$$

where $d_\alpha^{ij}$ is the element of the inverse matrix of the interdiffusion matrix. For calculating the solid-state transformation, the elastic driving force for the evolution of the phase-field variable is expressed as

$$\Delta G_{\alpha\beta}^{elastic} = -C_{ijkl}\varepsilon_{kl}^{el}\varepsilon_{ij}^0, \quad (16)$$

where $C_{ijkl}$ is the effective elastic modulus tensor, $\varepsilon_{ij}^{el}$ represents the elastic strain, and $\varepsilon_{ij}^0$ represents the eigenstrain. The effective elastic modulus is assumed to be uniform for the $\gamma$ and $\gamma'$ phases; that is, the elastic homogeneous case is considered for simplicity. The $\gamma$ and $\gamma'$ phases with four different variants are represented by five phase-field variables. The eigenstrain is assumed to be proportional to the local volume fraction of the $\gamma'$ phase:

$$\varepsilon_{ij}^0 = \varepsilon_0\delta_{ij}\left(1 - \phi_\gamma\right), \quad (17)$$

where $\varepsilon_0$ represents the lattice misfit between phases $\gamma$ and $\gamma'$, $\delta_{kl}$ represents the Kronecker delta function, and $\phi_\gamma$ is the phase-field variable of phase $\gamma$. The elastic strain is expressed as

$$\varepsilon_{kl}^{el} = \bar{\varepsilon}_{ij}^c + \delta\varepsilon_{ij} + \varepsilon_{ij}^0, \quad (18)$$

where $\bar{\varepsilon}_{ij}^c$ represents the homogeneous strain and $\delta\varepsilon_{ij}$ represents the heterogeneous strain. The homogeneous strain is calculated by averaging the eigenstrain over the volume $V$ as follows:

$$\bar{\varepsilon}_{ij}^c = \frac{1}{V}\int_V \varepsilon_{ij}^0 dV. \quad (19)$$

The heterogeneous strain is calculated from the mechanical equilibrium condition as follows:

$$C_{ijkl}\frac{\partial^2 u_k}{\partial x_j \partial x_l} = C_{ijkl}\frac{\partial \varepsilon_{kl}^0}{\partial x_j}, \quad (20)$$

where $u_k$ represents the local displacement.

### Computing environment

The simulation code is developed using the Taichi programming language—a domain-specific language integrated within Python[43]—to increase the computational efficiency. The Fast Fourier Transform in Numpy is used to solve the mechanical equilibrium equation. The Python version used in this study is 3.11, with Taichi version 1.7.2 and NumPy version 1.26.4. All figures were generated using matplotlib version 3.8.4. Simulations are conducted using a MacBook Pro laptop, Sonoma 14.5 OS, and an 11-core CPU with an Apple M3 Pro chip and 18 GB of unified memory.

### CALPHAD coupling method

A thermodynamic database is obtained from the MatCalc open database (https://www.matcalc.at/). This database is provided in the TDB format and had to be converted for utilisation in the phase-field simulations. The Python function for the chemical free energy is created using Python's regular expression operation library, Re. The contribution of the magnetic effects to the chemical free energy is neglected because it was very small, except at the Curie point. The first and second derivatives of the free energy is calculated using Taichi's differentiable programming and a Python function developed using

Python's analytical differentiation library Sympy, respectively. Although the analytical differentiation of Sympy can be used to develop a Python function for the first derivative of the free energy, differentiable programming is used because the CALPHAD functions become very long when differentiated once, making the Taichi compilation time-consuming. However, we use Sympy for the second derivative because CALPHAD functions become shorter when differentiated twice.

The system for converting TDB files into Python functions has been applied to TDB files released by the National Institute for Materials Science. This is called the Python module project. The converted Python functions are available from CPDDB.

### Reporting summary

Further information on research design is available in the Nature Portfolio Reporting Summary linked to this article.

### Data availability

The source data in the form of a Python file are also available on GitHub (https://github.com/takumimorino/phase-field). Source data are provided in this paper.

### Code availability

All the simulation codes, including the phase-field simulation, material parameters, and CALPHAD coupling (TDB conversion) codes, can be downloaded from GitHub (https://github.com/takumimorino/phase-field). The simulation codes are also accessible via Code Ocean[44] (https://codeocean.com/capsule/3744764/tree/v1). All codes are released under the MIT license.

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

## Acknowledgements

We would like to thank Niterra Materials Co., Ltd. (formerly Toshiba Materials Co., Ltd., Kanagawa, Japan) and JSPS KAKENHI Grant No. 22K04794 for financial support and Editage (www.editage.jp.) for English language editing.

## Author contributions

T.M. designed the research and performed coding and calculations. T.M. and M.O. wrote the paper. T.M., M.O. and S.H. contributed to the discussion and revision of this paper.

## Competing interests

The authors declare no competing interests.
