## [Transparent Peer Review file · Nature Communications]

An explicit integration approach for predicting the microstructures of multicomponent alloys

Corresponding Author: Mr Takumi Morino

Version 0:

Reviewer comments:

Reviewer #2

(Remarks to the Author)

The manuscript presents an extension of the classical KKS model to multicomponent, multiphase transformation in technical alloys. In particular, the “explicit formulation” of the CALPHAD minimization procedure is highlighted. This is interesting, but the realization is not new at all. Basically it hoys down to the “grand potential” approach, where the equilibrium chemical potential assumption is automatically fulfilled. (See W. Losert et al., Phys. Rev. E 58, 7492 (1998) R. Folch, M. Plapp, Phys. Rev. E 72, 011602 (2005), and there is work from Choudhury and Nestler Phys. Rev. E 85, 021602 and others) Now, if you truncate the expansion of the Gibbs energy to second order, the chemical potential becomes linear in composition. Then everything is “easy”. There is also numerous work on parabolic expansion of the Gibbs energy function.

For resubmission, please consider several items:

First of all: your approach is numerically efficient for isothermal simulations where, during the simulation no re-linearization of the chemical potential is needed. This has to be commented.

A general comment to KKS and extensions: If you set the interface to equilibrium, the driving force for phase transformation is gone. Please comment. Also KKS is not applicable for mixed mode transformations.

There derivation of the model equations is difficult to follow. In particular, I do not understand, how you derive eq. (6) from $H^I = 0$?

(Remarks on code availability)

Reviewer #3

(Remarks to the Author)

The authors present an explicit integration approach for a sublattice KKS model as an efficient method for modeling many component systems.

The idea of reformulating problems with internal iterative solves into explicit non-iterative schemes can be traced back more than 30 years with a 1991 paper by Nemat-Nasser being an example for an explicit plasticity solve. The application to phase field however is novel to my knowledge.

I would advise the authors to scrub the sensationalist sounding performance comparisons (“2.3 years” for pycalphad vs 293s or 63s with the authors’ approach). Those are apples to oranges (are you really comparing a single core python code to a parallel spectral solver?).

I'd also like to point out that everything from L121 through L150 is derived in reference [22] (Schwen et. al.), including the Lagrange multiplier approach. Eq. 3 is eq 8 in [22], eq 4 is eq 11,12 in [22]. Why not just cite that paper properly and focus on your contribution? This would also provide room to pull some parts of the derivation in the supplemental material into the main paper to highlight its core contribution.

L154: states that conventional simulations treat order parameters explicitly. Is that implying that a fully implicit solve is “unconventional”? I hope not, as that is a pretty standard approach.

The authors state that all terms on the right-hand side of eq. 7 are known. This requires a bit more explanation about the proposed solution method. The time derivative of the phase order parameters is most likely the result of a staggered Allen-Cahn / Cahn-Hilliard solve. This should be elaborated on.

The paper makes some absolute claims without proper justification, the wording should be reconsidered.

- "The PFM is the most effective method for analyzing the microstructures of materials." (line 94)
- "Two main thermodynamic models are used in CALPHAD: the quasi-regular solution model, which is primarily for liquid and solid solution phases, and the sublattice model for ordered phases, such as intermetallic compounds" -> Many other thermodynamic models exist and are widely used in CALPHAD, maybe specify that these are two the paper is focused on instead.

Fig. 1: The figure color map needs a legend. The caption should provide an explanation of what simulation the results are showing (is its initial condition or final, how much time did the system evolve). A figure should always be a self-sufficient explanation of what information is being conveyed in it.

Overall this is a nice contribution to the field and should be published. I appreciate the neat way of sharing the python implementation with reproducible runs. I know that I'll be implementing this method in my phase field solver (with the proper credit of course).

(Remarks on code availability)

The share code repository provides sufficient instructions (in the form of a Dockerfile). I was able to download and run the code using a python venv. The code looks neat and has plenty of comments that make it easy to follow.

Reviewer #4

(Remarks to the Author)

(Remarks on code availability)

Code can be easily downloaded and executed, or run directly through the CodeOcean capsule. This provides for excellent reproducibility.

Version 1:

Reviewer comments:

Reviewer #2

(Remarks to the Author)

Dear Authors,

thanks for the explanations. Now I think, that I have understood your approach. Interesting indeed. As I understand, it is based on Steinbach's finite interface dissipation model, which treats the phase compositions as independent variables, and relates the redistribution flux by a term which automatically sets the system into local quasi-equilibrium.

Some comments: I miss a statement, that you need to start from an equilibrium calculation in each point to set the initial phase concentrations. Second, as you know, it is dangerous to propagate a system explicitly, where you keep the initial flux as constant time step criterion. Then, from time to time you have to do an equilibrium calculation in order to check how much the system has drifted away. Please comment.

Some suggestions: Your writing is hardly intelligible, at least to me. You may use my description above, if you agree, to make clearer what you really do. The word "quasi-equilibrium", which you promise in the rebut, does not appear in the text. Please comment also on "mixed mode" transformation, where the quasi-equilibrium condition is violated.

I find the Maclaurin expansion superficial. If you hard code a regular solution model, then everything is known. If you use a commercial, encrypted database, your approach is not feasible

(Remarks on code availability)

Well done, but rather basic.

Reviewer #3

(Remarks to the Author)

Thank you for addressing my concerns. The revised document is fit for publication.

(Remarks on code availability)

Reviewer #4

(Remarks to the Author)

(Remarks on code availability)

Code runs out of the box and is easily accessible through the codeocean link.

Version 2:

Reviewer comments:

Reviewer #2

(Remarks to the Author)

All corrections well done, thank you, ready for publication from my side.

(Remarks on code availability)

Report of the First Referee

Comment 1.

Basically it boils down to the “grand potential” approach, where the equilibrium chemical potential assumption is automatically fulfilled. (See W. Losert et al., Phys. Rev. E 58, 7492 (1998) R. Folch, M. Plapp, Phys. Rev. E 72, 011602 (2005), and there is work from Choudhury and Nestler Phys. Rev. E 85, 021602 and others) Now, if you truncate the expansion of the Gibbs energy to second order, the chemical potential becomes linear in composition. Then everything is “easy”. There is also numerous work on parabolic expansion of the Gibbs energy function.

Response:

We apologize for any misunderstanding our explanation may have caused. Our model differs fundamentally from the grand potential approach. In the grand potential approach, a conversion between composition and chemical potential is required, necessitating an explicit relationship between them. Thus, it is necessary to approximate or linearize the Gibbs free energy function; i.e., direct use of the CALPHAD functions is challenging. Modifying the functional form of the free energy in this manner can lead to non-quantitative results, even if the grand potential model itself is formulated quantitatively. To overcome this limitation, we developed our model to enable the direct use of any free-energy function, even when the relationship between composition and chemical potential is not explicitly defined. This is possible because the time evolution equation for the site fraction in our model is an explicit function and no conversion between composition and chemical potential is required.

We suspect that our use of a Maclaurin expansion when deriving the evolution equation of the site fraction may have given the impression that we neglected the higher order in the Gibbs energy. However, while we employed a Maclaurin expansion in the derivation process, the free-energy functions used in the simulations were not modified in any way. As a result, our model ensures that the equal diffusion potential and internal equilibrium conditions are satisfied using the original CALPHAD free functions, without any alterations, such as linearization and parabolic expansion. In contrast, the grand potential model inherently requires modifying the free-energy function.

We also believe that our approach can be extended to the grand potential framework, allowing calculations with arbitrary free-energy functions, and we plan to address this in future work.

Changes:

- Lines 78–83: We have described the relationships between the present model and the grad potential approach as follows:

“In the grand potential approach^{20,21}, the equal diffusion potential condition is automatically fulfilled; however, it is necessary to express the chemical potential explicitly as a function of composition. This involves linearizing the chemical potential or using parabolic approximations of free-energy functions with respect to composition, which makes the direct use of CALPHAD functions challenging. To address this problem,”

- Lines 213–216: We have revised the explanation of Fig. 1 from

“As these simulations utilize the CALPHAD database, Fig. 1 depicts the partitioning of each element into phases, which is informed by a phase diagram.”

to

“As these simulations directly utilize the CALPHAD database without simplifications for computational speed, Fig. 1 shows the partitioning of each element into phases based on a phase diagram, ensuring accurate and reliable results.”

- Lines 266–268: We have added an explanation that the present model can be directly integrated with the CALPHAD database:

“The proposed approach can be directly integrated with the CALPHAD database without modifying functional forms.”

Comment 2.

First of all: your approach is numerically efficient for isothermal simulations where, during the simulation no re-linearization of the chemical potential is needed. This has to be commented.

Response:

Thank you for your comment. Our approach is numerically efficient for both isothermal and non-isothermal simulations. As we addressed in the response to Comment 1, our model allows the direct use of the CALPHAD functions with the chemical potential expressed as a function of composition and temperature, without linearization of the chemical potential or parabolic approximations of free-energy functions. In fact, the simulations of solidification presented in our paper are non-isothermal.

Changes:

- Lines 198–202: We have added an explanation that the present model is numerically efficient for both isothermal and non-isothermal simulations:

“Note that the proposed model directly employs CALPHAD functions without the need for modifications, such as linearizing the chemical potential or applying parabolic approximations to free-energy functions. This approach allows efficient computation regardless of temporal and spatial temperature changes, including those occurring during processes such as alloy solidification.”

Comment 3.

A general comment to KKS and extensions: If you set the interface to equilibrium, the driving force for phase transformation is gone. Please comment. Also KKS is not applicable for mixed mode transformations.

Response:

Thank you for your insightful comment. We believe that it is important to clarify what we mean by “equilibrium at the interface.” In the KKS model, the driving force for phase transformation is calculated as the difference of “free energy minus the composition multiplied by the diffusion potential” for each phase—essentially, the difference in the grand potential for each phase. Therefore, the interface is not in a complete thermodynamic equilibrium (common tangent) but rather in a quasi-equilibrium state (parallel tangent).

Additionally, by using the thin interface limit and antitrapping approaches for systems with arbitrary values of the diffusivities and interfacial energies [M. Ohno and K. Matsuura, Phys. Rev. E 79, 031603 (2009) M. Ohno, T. Takaki, and Y. Shibuta, Phys. Rev. E 96, 033311 (2017)], it is possible to compute mixed-mode transformations. Setting the interface mobility to infinity allows calculations of diffusion-limited transformations, while setting it to a finite small value allows for calculations of reaction-limited transformations.

Changes:

- Lines 400–402: We have added an explanation that the driving force for interface motion is calculated according to the quasi-equilibrium assumption:

“The driving force for interface motion due to chemical free energy is calculated from the distance between the parallel tangents to the free-energy curves of the two phases, assuming that the interface is in quasi-equilibrium^{7,17}.”

Comment 4.

There derivation of the model equations is difficult to follow. In particular, I do not understand, how you derive eq. (6) from $H^I = 0$?

Response:

We apologize for the lack of clarity. A detailed derivation of Eq. (6) from $H^I = 0$ is provided in the Supplementary information, and we hope that this will clarify the process.

Changes:

- Lines 172–173: We have added “(see Supplementary Information for details)”.
- Lines 177–178: We have added “(see Supplementary Information for details)”.

Report of the Second Referee

Comment 1.

I would advise the authors to scrub the sensationalist sounding performance comparisons (“2.3 years” for pycalphad vs 293s or 63s with the authors’ approach). Those are apples to oranges (are you really comparing a single core python code to a parallel spectral solver?).

Response:

Thank you for your valuable comment. You are correct, and we apologize for the confusion. The phase-field calculations were accelerated using Taichi, while the pycalphad calculations were not similarly optimized. Therefore, a direct comparison is not entirely appropriate.

Changes:

- Line 245: We have removed the following statement:
“, which would take 2.3 years with PyCalphad”

Comment 2.

I’d also like to point out that everything from 1121 through 1150 is derived in reference [22] (Schwen et. al.), including the Lagrange multiplier approach. Eq. 3 is eq 8 in [22],

eq 4 is eq 11,12 in [22]. Why not just cite that paper properly and focus on your contribution? This would also provide room to pull some parts of the derivation in the supplemental material into the main paper to highlight its core contribution.

Response:

You are correct that the derivations on lines 121 to 150 largely follow those in Schwen et al., and we should have appropriately cited their work. We apologize for this oversight and appreciate your constructive feedback.

Changes:

- Line 116: We have changed the subheading from
“Local minimization condition”
to
“Definition of variables and local minimization condition”.
- Lines 118–121: We have shorted the sentence by citing the work of Schwen et al.
- Line 141: We have cited the paper of Schwen et al.
- Line 144: We have cited the paper of Schwen et al.

Comment 3.

L154: states that conventional simulations treat order parameters explicitly. Is that implying that a fully implicit solve is “unconventional”? I hope not, as that is a pretty standard approach.

Response:

Thank you for pointing out that the wording may have been misleading.

Changes:

- Line 154: We have removed the following text:
“the composition and phase-field variables are handled explicitly, whereas”

Comment 4.

The paper makes some absolute claims without proper justification, the wording should be reconsidered.

- “The PFM is the most effective method for analyzing the microstructures of materials.”
(line 94)

- “Two main thermodynamic models are used in CALPHAD: the quasi-regular solution model, which is primarily for liquid and solid solution phases, and the sublattice model for ordered phases, such as intermetallic compounds” -> Many other thermodynamic models exist and are widely used in CALPHAD, maybe specify that these are two the paper is focused on instead.

Response:

As you correctly mentioned, the statements required proper justification.

Changes:

- Lines 47–48: We have changed
“Two main thermodynamic models are used in CALPHAD”
to
“This paper focuses on two thermodynamic models”.
- Line 98: We have changed
“The PFM is the most effective method for”
to
“The PFM is widely used for”.
- Line 98: We have cited phase-field papers ([3]–[5]).

Comment 5.

Fig. 1: The figure color map needs a legend. The caption should provide an explanation of what simulation the results are showing (is its initial condition or final, how much time did the system evolve). A figure should always be a self-sufficient explanation of what information is being conveyed in it.

Response:

Thank you for your helpful comment regarding the figure captions. We have made the following modifications to help readers better understand the practicality of the proposed method.

Changes:

- Fig. 1: We have added a scale bar and description of phases.
- Fig. 1: We have removed Figs. 1a and b because we believe that Fig. 1c, which has the largest number of alloying elements, is sufficient to support the main argument of the paper. We consider that this revision improved the clarity of the manuscript.
- Fig. 1: We have changed the range of colormap to emphasize the difference in the compositional distribution among each elements.

- Lines 203–208: We have changed the caption to explain what the simulation results show (which timing the result corresponds to, for how much time the system evolved), from

“Fig. 1 | Composition distributions of the simulated microstructures. a, Solidification in the 12-component Al system. b, Solidification in the 15-component Ni system. c, Solidification in the 20-component Fe system. d, Solid-state transformation in the 12-component Ni system.”

to

“Fig. 1 | **Snapshots of composition distributions at the end of the two simulations.** (a) Solidification in a 20-component Fe system at a solidification time of 6.0×10^{-2} s. (b) Solid-state transformation in a 12-component Ni system at an aging time of 1.5×10^3 s. The range of the colormaps is based on the average composition of each element (c_{ave}). Owing to the direct use of the CALPHAD database, the partitioning of each element varies according to the phase diagram.”

- Lines 217–219: We have added an explanation that some figures are omitted:

“However, the figures for these systems are omitted here, as they exhibit similar patterns of concentration distribution, confirming the robustness of our computational method across different alloy compositions.”

Active correction

Correction 1.

The system for converting TDB files into Python functions, as described in the “CALPHAD coupling method (Line 448)” has been applied to TDB files released by the National Institute for Materials Science. This is called the “python module project.” The converted Python functions are available from CPDDB (<https://cpddb.nims.go.jp/>).

Changes:

- Lines 461–463: We have added an explanation of the “python module project”:

“The system for converting TDB files into Python functions has been applied to TDB files released by National Institute for Materials Science. This is called the “python module project.” The converted Python functions are available from CPDDB (<https://cpddb.nims.go.jp/>).”

- Lines 493–504: We have added Acknowledgements, Author contributions, Competing interests and Correspondence.

Report of the Reviewer #2

Comment 1.

I miss a statement, that you need to start from a equilibrium calculation in each point to set the initial phase concentrations.

Response:

Thank you for your comment. We realize that the explanation of the computational procedure was insufficient. The computational process began with (i) setting the initial conditions for temperature, composition, and phase fields. The initial site fraction (or phase composition) was determined by equilibrium calculations using CALPHAD software pycalphad. Subsequently, (ii) the governing equation of the phase-field variable and composition are solved. Following this, (iii) the governing equation of site fraction represented by Eq. (7) is addressed. It should be noted that the phase compositions are automatically determined through their relationship with the site fraction, as described in Eq. (2). Finally, (iv) steps (ii) and (iii) are iteratively repeated.

Changes:

- Lines 204-210: We have added an explanation about computational procedure as follows:

“The computational process began with (i) setting the initial conditions for temperature, composition, and phase fields. The initial site fraction was input as the equilibrium values corresponding to these conditions. Subsequently, (ii) the governing equations of the phase-field variable and composition are solved. Following this, (iii) the governing equation of site fraction represented by Eq. (7) is addressed. It should be noted that the phase compositions are automatically determined through their relationship with the site fraction, as described in Eq. (2). Finally, (iv) steps (ii) and (iii) are iteratively repeated.”

Comment 2.

Second, as you know, it is dangerous to propagate a system explicitly, where you keep the initial flux as constant à time step criterion. Then, from time to time you have to do

an equilibrium calculation in order to check how much the system has drifted away. Please comment.

Response:

Thank you for your comment. The computation will not become unstable even when using a constant time step because in deriving Eq. (7), numerical stability condition was considered. A detailed discussion of the stability condition has been added to the Supplementary Information.

Even though the system propagates explicitly, it does not drift away because the first term in the numerator of Eq. (7) corresponds to the redistribution flux in the Steinbach's finite interface dissipation model. This term adjusts the site fraction (and phase composition) to meet the local minimization condition (i.e., equal diffusion potential and internal equilibrium conditions). Figure 2 shows the error in the local minimization condition evaluated at all grids points for every time step. The error was negligibly small, which confirms that the local minimization condition is consistently satisfied throughout the simulation and that no drift occurs.

Changes:

- (Supplementary information) Lines 68-96: We have added a detailed discussion of the stability condition.
- Lines 180 to 181 : We have added an explanation that the calculation does not drift away as follows:

“The first term in the numerator of Eq. (7) adjusts the site fraction to meet the local minimization condition.”
- Lines 249 to 250 : We have added an explanation that the calculation does not drift away as follows:

“Therefore, it was confirmed that the local minimization condition is maintained throughout the simulation.”

Comment 3.

The word “quasi-equilibrium”, which you promise in the rebut, does not appear in the

text. Please comment also on “mixed mode” transformation, where the quasi-equilibrium condition is violated.

Response:

We apologize for the confusion and our mistake. The present model is not applicable to cases where the quasi-equilibrium condition is violated such as in mixed-mode transformations.

Changes:

- Lines 416-417: We have added an explanation that the present model is not applicable to cases where the quasi-equilibrium condition breaks down as follows: “Therefore, the present model is not applicable to cases where the quasi-equilibrium condition is violated such as in mixed-mode transformations.”

Comment 4.

I find the Maclaurin expansion superficial. If you hard code a regular solution model, then everything is known. If you use a commercial, encrypted database, your approach is not feasible

Response:

To calculate the site fraction, it is generally necessary to solve both the equal diffusion potential condition and internal equilibrium condition, which requires convergence calculation and is therefore computationally expensive. In lines 44-50 of Supplementary Information file, we performed a Maclaurin expansion and neglected higher-order terms of Δy , assuming that the change in the site fraction per time step is sufficiently small such that terms of $(\Delta y)^2$ and higher can be ignored. This expansion yields an explicit function of Δy represented by Eq. (7). Solving Eq. (7) ensures that the equal diffusion potential condition and internal equilibrium condition are satisfied without convergence calculations, thereby dramatically reducing the computational cost.

All CALPHAD data including those for the quasi-regular solution model and the sublattice model have been implemented directly in the phase-field simulation code. For ultra-multicomponent systems, such as those with 20 components, hard-coding manually becomes impractical; therefore, we developed an automatic code generation system. The details are described in “CALPHAD coupling method” section (Line 465). Even when using a commercial, encrypted database, our model can still be used for calculations by accessing the database through the software’s API.

Changes:

- (Supplementary information) Lines 47-48: We have added the rationale for performing the Maclaurin expansion and neglecting the higher-order terms as follows:

“The change in the site fraction per time step (Δy) is sufficiently small such that terms of $(\Delta y)^2$ and higher can be ignored.”
- Lines 181-183: We have explained the improvement in computational efficiency achieved by using Eq. (7) as follows:

“As a result, solving Eq. (7) alone satisfies the local minimization condition without the need for convergence calculations, which is expected to significantly reduce the computational cost.”
- Lines 210-216: We have described that all CALPHAD data are hard-coded as follows:

“All CALPHAD data including those for the quasi-regular solution model and the sublattice model have been implemented directly in the phase-field simulation code. For ultra-multicomponent systems, such as those with 20 components, manually hard-coding becomes impractical due to complexity; therefore, we developed an automatic code generation system as described in Method section. In the case of commercial, encrypted databases, the necessary values for calculations can be obtained through the software’s API.”